# Research

 

**Cite this article:** da Silva Coelho FA, Gill S, Tomlin CM, Heaton TH, Lindqvist C. 2021 An early dog from southeast Alaska supports a coastal route for the first dog migration into the Americas. *Proc. R. Soc. B* **288**: 20203103. https://doi.org/10.1098/rspb.2020.3103

evolution, genomics, palaeontology

*Canis lupus familiaris*, North Pacific Coast, palaeodiet, palaeogenetics, precontact dogs, southeast Alaska

**Author for correspondence:**
Charlotte Lindqvist
e-mail: cl243@buffalo.edu

[†]These authors contributed equally to this study.

# An early dog from southeast Alaska supports a coastal route for the first dog migration into the Americas

Flavio Augusto da Silva Coelho[1,†], Stephanie Gill[1,†], Crystal M. Tomlin[1], Timothy H. Heaton[2] and Charlotte Lindqvist[1,3]

[1]Department of Biological Sciences, University at Buffalo, Buffalo, NY 14260, USA
[2]Department of Earth Sciences, University of South Dakota, Vermillion, SD 57069, USA
[3]School of Biological Sciences, Nanyang Technological University, Singapore 637551, Singapore

(iD) FASC, 0000-0003-0236-7044; CL, 0000-0002-4190-727X

The oldest confirmed remains of domestic dogs in North America are from mid-continent archaeological sites dated approximately 9900 calibrated years before present (cal BP). Although this date suggests that dogs may not have arrived alongside the first Native Americans, the timing and routes for the entrance of New World dogs remain uncertain. Here, we present a complete mitochondrial genome of a dog from southeast Alaska, dated to $10\,150 \pm 260$ cal BP. We compared this high-coverage genome with data from modern dog breeds, historical Arctic dogs and American pre-contact dogs (PCDs) from before European arrival. Our analyses demonstrate that the ancient dog belongs to the PCD lineage, which diverged from Siberian dogs around 16 700 years ago. This timing roughly coincides with the minimum suggested date for the opening of the North Pacific coastal (NPC) route along the Cordilleran Ice Sheet and genetic evidence for the initial peopling of the Americas. This ancient southeast Alaskan dog occupies an early branching position within the PCD clade, indicating it represents a close relative of the earliest PCDs that were brought alongside people migrating from eastern Beringia southward along the NPC to the rest of the Americas. The stable isotope $\delta^{13}C$ value of this early dog indicates a marine diet, different from the younger mid-continent PCDs' terrestrial diet. Although PCDs were largely replaced by modern European dog breeds, our results indicate that their population decline started approximately 2000 years BP, coinciding with the expansion of Inuit peoples, who are associated with traditional sled-dog culture. Our findings suggest that dogs formed part of the initial human habitation of the New World, and provide insights into their replacement by both Arctic and European lineages.

## 1. Introduction

Dogs have followed humans across the globe since their domestication [1,2] and been an integral part of human colonization of the New World. However, the exact timing and route(s) of human migration into the Americas remain as unresolved as those of their canine companions. While genetic evidence suggests that the first Native Americans split from East Asian ancestors as early as 23 000 years ago [3–6], and evidence from archaeological artefacts indicates a presence of human populations south of the continental ice sheets at possibly more than 16 000 years ago [7], the earliest human remains in North America date to only around 12 600 years ago [8]. Genetic data for dogs indicate that their Old World domestication process occurred between 32 100 and 18 800 years ago [9], early enough to accompany the first North American immigrants. However, as with humans, archaeological evidence for dogs lags behind DNA-based dates, at no earlier than 15 000–12 500 years ago in the Old World [10].

The gaps between genetic and physical evidence leave open the question of whether dogs accompanied humans during the initial peopling of the Americas, or if dogs arrived alongside humans only in later migration waves [11,12]. So far, the oldest confirmed domestic dog remains in North America, excavated from dog burials at the Koster and Stilwell II archaeological sites in Illinois, date to between 10 190 and 9630 calibrated years before present (cal BP) [13]—more than 2000 years younger than the oldest confirmed human remains. A recent genetic study demonstrated that these early pre-European contact dogs (PCDs) were not domesticated from North American wolves, having instead diverged from an ancestor shared with an approximately 9000-year-old dog population that inhabited Zhokhov Island in Eastern Siberia [11,14].

New World human and dog prehistory is also intimately dependent on the timing and extent of Ice Age glaciations, as well as biological productivity and food availability during and immediately following the Last Glacial Maximum (LGM). During the LGM, the Laurentide Ice Sheet and the Cordilleran Ice Sheet covered much of North America, making the area inhospitable or even impassible to humans and most other fauna and flora [5]. For many decades, it was accepted that early humans migrated through an ice-free corridor in between the two ice sheets that opened once deglaciation initiated [15–17]. Although this corridor may have opened as early as approximately 15 000 years ago [16], it was likely not biological viable until approximately 13 000 years ago [17–19]. Another hypothesis is that the retreat of the Cordilleran Ice Sheet along the Northwest Pacific coast (NPC), combined with ocean levels lower than today, provided an earlier, ice-free coastal corridor for early humans and other biota [20,21]. Islands along the NPC, including the Alexander Archipelago in southeast Alaska, held a unique biotic diversity and were resource-viable by approximately 17 000 ago, when the Cordilleran Ice Sheet started to retreat at its westernmost edge [20,22]. This timing is roughly contemporaneous with the genetic evidence for the first human migration wave into the New World [5,15,18,20], and consistent with genetic data that suggest dogs had already been domesticated. Whether dogs colonized North America with the earliest humans or only later, they spread broadly around the continent. Archaeological evidence shows that dogs were present at different mid-continental sites approximately 10 000 years ago [13], spanning from Newfoundland to Alabama and California by 4000 years ago, and into South America by 1000 years ago [9,11,23].

Modern and ancient dogs are grouped into four major haplogroups (A–D), and two minor haplogroups (E–F) (e.g. [24,25]). Haplogroup A is the most diverse, including most dog breeds [9] and the PCD subclade [11,25]. Over the last 2000 years, at least three dog migrations into the New World could have impacted the PCD population directly. First, Arctic dogs arrived from East Asia with the Thule culture [26]. Before the Thule culture, dogs were rare in the American Arctic. The rapid expansion of dogs across the North American Arctic was likely associated with the later Inuit dog sled culture [25–28]. Europeans brought a new wave of modern dog breeds, replacing most PCDs, mainly due to cultural preferences. Colonists preferred larger European breeds over the smaller PCDs because they were more capable of defending settlements and livestock, and they could be used as war dogs or for hunting [27,29]. European

dog breeds may have also brought diseases that indigenous dogs were susceptible to [11]. More recently, Siberian huskies were introduced during the Alaska Gold Rush [28].

Because the oldest New World dog remains are from the American Midwest, they do not provide strong insight into which route PCDs could have taken while migrating to the Americas. Direct evidence from ancient PCDs along the NPC would help solve this conundrum. Here we describe an ancient dog mitochondrial genome that dates to 10 150 ± 260 cal BP, excavated from Lawyer's Cave on the Alaskan mainland east of Wrangell Island in the Alexander Archipelago of southeast Alaska. Known as PP-00128, it represents the oldest bone remains found in that cave, and is the oldest genetically confirmed dog discovered in the New World.

## 2. Material and methods

### (a) Lawyer's Cave and specimen description

The PP-00128 specimen (University of Alaska Museum Earth Sciences Collection, Fairbanks, catalogue number UAMES 52 399) was discovered in Lawyer's Cave, also called Phalanges Phreatic Tube by cavers, referring to the cave's shape and the toe bones of a bear found inside. The cave, which has two entrances and consists of an approximately 20 m long, non-branching crawlway from end to end, is located along Blake Channel on the southeast Alaskan mainland east of Wrangell Island in the Alexander Archipelago (figure 1). Lawyer's Cave is rich in other postglacial remains beyond the PP-00128 specimen, including bones of various mammals, birds and fish, as well as human remains and artefacts that were discovered during two excavations in 1998 and 2003 [30,31]. The latter consist of small human bones (likely from the same individual) and several artefacts (a bone spear point, bone awl, shell beads, partial obsidian biface, obsidian microblade and obsidian flakes). The bone awl has been dated to 3050 ± 40 [14]C years BP [30,31]. The PP-00128 bone remains are a piece of the dome-like head of a femur with a diameter of about 1 cm (figure 1). Initially, the bone fragment was identified as of uncertain mammalian origin but suspected to be from a bear (*Ursus*). However, as part of a larger genetic survey of remains from this cave (C. Lindqvist 2020, unpublished data), PP-00128 was identified as *Canis lupus*.

### (b) Radiocarbon dating and stable isotopes

Radiocarbon dating and stable isotope $\delta^{13}$C of PP-00128 was performed in 2004 by the geochronology laboratory at the University of Arizona (AA-56999; see Heaton & Grady [31] for details). The [14]C date was calibrated using the IntCal13 calibration curve [32] in OxCal v4.3 [33], with 2-sigma dates reported. Stable isotope $\delta^{13}$C can be used as a proxy to analyse palaeodietary patterns [34], and the $\delta^{13}$C value from PP-00128 was compared to other dogs (Arctic and modern dogs), and mammals, as well as salmon, an abundant food source present in the region.

### (c) DNA extraction, PCR amplification, mitochondrial enrichment and mitogenome mapping

Genomic DNA was extracted in a dedicated cleanroom facility appropriate for ancient DNA research, physically separated from any handling of modern samples and post-PCR procedures. The ancient DNA extraction followed the protocol described in Dabney *et al.* [35], with some modifications. Using a dentist drill, approximately 100 mg of fine bone powder was obtained. Following overnight digestion with proteinase K and addition of the binding buffer, the mixture was purified and concentrated with a Qiagen MinElute PCR Purification Kit (Qiagen, USA). The

Proc. R. Soc. B 288: 20203103

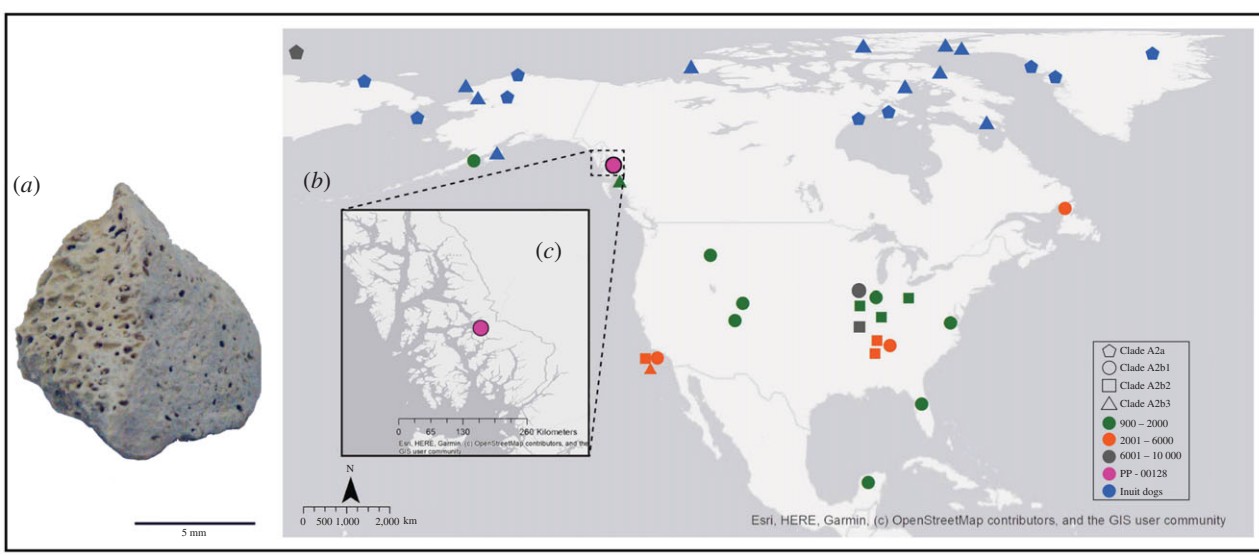

**Figure 1.** (*a*) The PP-00128 bone remains, which are a piece of the dome-like head of a femur. The scale represents 5 mm. (*b*) Map indicating locations of remains for each ancient mitogenome used in this study of precontact dogs, Arctic sled dog and Zhokov dogs. The shapes of the dog samples represent subclade affinity (figure 2*b*) and colours correspond to three age classes and Inuit dog locations (see legend). The location of the southeast Alaskan dog specimen, PP-00128, is coloured in pink. (*c*) Map of the Alexander Archipelago, indicating the location of Lawyer's Cave (pink circle), where the PP-00128 remains were found. (Online version in colour.)

final elution step was performed twice with 25 µl of TE buffer for a total DNA volume of 50 µl. A negative control was prepared alongside the extraction.

As part of initial screening for species identity, PCR reactions were prepared in the cleanroom by adding 21 ml H$_2$O, 1 ml of each forward and reverse primer, and 2 ml genomic DNA to each GE illustra PuReTaq Ready-To-Go PCR bead (GE health-care, USA). Because the sample was initially suspected to be bear, two primer pairs designed for the amplification of the *Ursus* control region and cyt*b* were used (primers 161F/162R & 164R/165F, respectively; [36]). A touchdown thermal cycling pro-tocol was used as follows: 10 min at 94°C, 10 cycles of 30 s at 94°C, 30 s annealing with the temperature decreasing every cycle by 0.5°C from 55°C to 50°C, and 30 s extension at 72°C, followed by 25 cycles with annealing temperature set to 50°C and denatura-tion and extension phases as above. One primer pair (164R/165F) successfully amplified and was Sanger sequenced using the same primer pair.

For whole mitochondrial genome sequencing, extracted DNA was prepared for Ion Torrent sequencing and mtDNA target enrichment by Arbor Biosciences (http://www.arbor-biosci.com). The Ion Torrent sequencing library was target-enriched using a custom-designed ursid mitogenome bait panel manufactured by Arbor Biosciences [36] following the standard MYbaits v. 3.0 protocol. Equal masses of libraries were pooled, bead-templated, and sequenced on the Ion Proton platform using the Ion PI Chip Kit v2 chemistry. Following sequencing, reads were de-multiplexed, quality-trimmed, and filtered using the default settings on the Ion Torrent Suite v. 4.4.3. Another round of mtDNA enrichment and sequencing was performed based on a domestic dog mitogenome bait panel (designed from NCBI accession NC002008.4). An Illumina Truseq dual-barcoded library was prepared without sonication, using the blunt-end ligation module from the NEBNext Fast DNA library preparation kit (New England Biolabs, E6270 L) with an extended double-time treatment and blunt-end adapters syn-thesized by Arbor Biosciences. The library was target-enriched using the Arbor Biosciences mitogenome panel and paired-end sequenced on the Illumina NovaSeq platform. Illumina reads were trimmed using Trimmomatic [37].

All Ion Torrent and Illumina reads were aligned separately against the CanFam 3.1 mitogenome using BWA v. 0.7.13 [38]

with the aln algorithm and subsequence seed set to 1024. Illu-mina read pairs were mapped separately and then merged. Unmapped reads were extracted with Samtools [39] and then mapped with BWA-mem [40] using default parameters. PCR duplicates were removed with the MarkDuplicates tool in the Picard software suite v. 1.112 using lenient validation stringency (http://broadinstitute.github.io/picard/). Consensus calling was carried out using Samtools mpileup [41] with default set-tings. To obtain a better assembly, the individual BAM files generated from mappings of IonTorrent and Illumina reads, respectively, were merged using Picard MergeSamFiles. A con-sensus sequence from this merged assembly was also generated using Samtools mpileup [41]. Mapping statistics were calculated with bedtools [42].

## (d) DNA degradation assessment

Ancient DNA has a high rate of nucleotide misincorporation due to post-mortem damage. These misincorporations, mostly caused by deamination, follow a well-known degradation pattern that is used to assess aDNA authenticity. We used MapDamage 2.0 [41], which uses an approximate Bayesian estimation of the damage patterns to assess DNA degradation patterns in the reference mapped assembly of PP-00128. We also used Schmutzi [43] to obtain an endogenous mitogenome sequence, putatively free of modern human (or other) contamination. Schmutzi esti-mates contamination in the genome based on deamination patterns and fragment length distribution to reconstruct an endogenous mitochondrial genome sequence. The recon-structed endogenous mitogenome was used for downstream analyses.

## (e) Mitogenome sequence data analysis

Complete mitochondrial genomes of modern dog breeds, ancient dogs, as well as ancient and modern wolves were downloaded from NCBI GenBank and the European Nucleotides Archive (ENA). Four distinct datasets were constructed, all containing sequences from modern and ancient dogs, representing different clades and countries, ancient and modern wolves, previously sequenced precontact dogs (figure 1), and the newly generated sequence from PP-00128. The first dataset comprised 1208

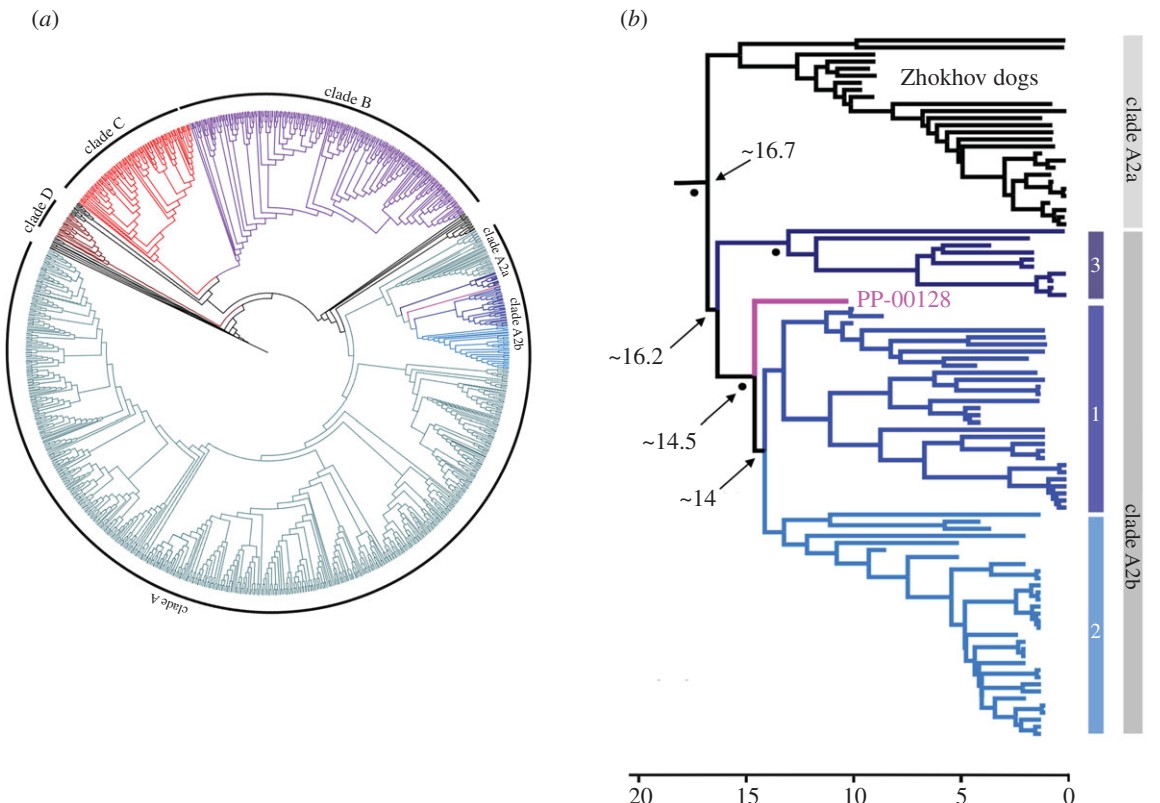

**Figure 2.** A tip-calibrated Bayesian phylogenetic tree of 1106 dog and wolf mitogenomes (see complete tree with tip labels in electronic supplementary material figure S8). (*a*) The complete tree showing the four main dog haplogroups (A–D), and subclades A2a and A2b. (*b*) The same phylogeny but showing only the subclades A2a and A2b, including the precontact dog subclades A2b1, A2b2 and A2b3 discussed in the text. Black dots indicate posterior probability higher than 0.95. Arrow with numbers pointing to selected nodes indicates divergence time estimates (in thousands of years). (Online version in colour.)

sequences (dataset 1208), including mitogenome sequences from three previous studies [10,11,25]. The second dataset had 1106 sequences (dataset 1106), only retaining sequences with a minimum coverage of 10X. The third dataset had 1099 sequences (dataset 1099), composed of sequences similar to dataset 1106, but excluding modern dogs that group inside the PCD clade in the study of Ní Leathlobhair *et al.* [11]. The final and fourth dataset comprised 1048 sequences (dataset 1048), including only sequences from [10,11], but excluding the Arctic sled dog sequences from [25]. Alignments of the sequences were performed in MAFFT [44], followed by manual adjustment in Geneious Prime 2020.2.1 (https://www.geneious.com) to exclude the variably numbered tandem repeats of the D-loop. Additionally, 59 base pairs from the beginning and 66 from the end of the alignment were trimmed to make all sequences the same length. Maximum-likelihood phylogenetic analysis of all datasets was performed using RAxML-HPC Black Box 8.2.12 [45], subsampled with 1000 bootstraps, in the CIPRES Science Gateway [46] under the GTR substitution model, which was chosen as the best-supported model by ModelTest-NG 0.1.5 [47]. To obtain estimated divergence times, a BEAST v. 1.10.4 [48] analysis was performed only with datasets that contained sequence data with a minimum depth coverage of 10X (hence, it was not performed on dataset 1208). The GTR substitution model, a strict clock, and a constant-size coalescent model were used. Age calibration of the terminal nodes was done using the calibrated radiocarbon dates from the radiocarbon-dated samples. Trees were sampled every 5000 states from a total of 140 million states. TreeAnnotator was used to generate a maximum clade credibility tree, with a 30% burn-in. Effective sampling size (ESS) values were higher than 200 for all parameters sampled from the MCMC, and the posterior distributions were examined using Tracer v. 1.7.1 [49]. In order to investigate the demographic history of precontact dogs, an

Extended Skyline Plot (ESP), using a dataset composed only of ancient dogs from the A2b clade and sequences with a minimum coverage of 10X, was performed using BEAST2 [50] under the GTR model with 100 million iterations. A median-joining haplotype network of sequences from precontact dogs and Zhokov dogs was generated using PopArt [51].

# 3. Results

## (a) Radiocarbon calibration and stable isotope $\delta^{13}C$

The radiocarbon dating of PP-00128 returned a date of 9020 ± 85 $^{14}C$ years BP, with a calibrated date ranging from 10 412 to 9888 2-sigma cal years BP and a median date of 10 150 ± 262 years BP. Hence, this date is slightly older, by about 240 years, than the to-date earliest confirmed New World domestic dog remains, which are dated to 8840 ± 80 $^{14}C$ years BP with a calibrated median date of 9910 ± 280 years BP [13].

PP-00128 had a $\delta^{13}C$ value equal to −14.1‰. Compared to data from numerous terrestrial and marine mammals [13,52–60] and salmon (a common food resource in the region where PP-00128 was found), PP-00128 has a $\delta^{13}C$ at the higher end of the $\delta^{13}C$ spectrum, similar to marine mammals, particularly polar bears (electronic supplementary material, figure S1). This result indicates that the diet of this ancient dog was predominantly marine, with a value similar to ancient dogs from Nunalleq, a precontact Yup'ik archaeological site in coastal Southwestern Alaska [54]. This is in contrast to other contemporaneous dogs that inhabited the Midwest region [13], as their values overlap with the lower end of the range for modern dogs and are within the range of non-carnivorous mammals.

## (b) Mitogenome mapping and DNA degradation

The mitogenome assembly of the Ion Torrent sequences had a coverage of 43X, and a width of 94%, while the assembly of the Illumina sequence data had a coverage of 138X and a width of 100%. After merging the BAM files from both methods, the coverage was 180X, with a width of 100%. The assembly QC statistics are presented in electronic supplementary material, table S1. As anticipated for an ancient sample, the map-damage analysis of PP-00128 mitogenome showed an increased rate of cytosine deamination at the 5′ end of the reads and an increase of guanine to adenine substitution close to the 3′ end of the reads (electronic supplementary material, figure S2). Schmutzi indicated no modern human contamination.

## (c) Phylogenetic analyses

Phylogenetic inference based on maximum-likelihood RAxML analysis was conducted on all four datasets: dataset 1208 (electronic supplementary material, figures S3 and S4), 1106 (electronic supplementary material, figure S5), 1099 (electronic supplementary material, figure S6) and 1048 (electronic supplementary material, figure S7), as well as Bayesian inference BEAST analysis for datasets 1106 (figure 2a), 1099 (electronic supplementary material, figure S9) and 1048 (electronic supplementary material, figure S10). In all trees analysed, the PP-00128 bone specimen groups among the PCDs, constituting a monophyletic group within haplogroup A and closely related to a dog population from Zhokov Island in Siberia. Ancient and modern dogs inhabiting the Arctic region of North America and Greenland also group with the Zhokov Island dog lineage, denoted haplogroup A2a. The PCD lineage, denoted haplogroup A2b, forms three separate subclades in our analyses (figures 1b and 2b): (1) A2b1 that includes the ancient dogs from the Koster site, dogs ranging in age from approximately 4000 to 1000 years old from throughout North America, California's Channel Islands and into South America, as well as a group of modern dog breeds, (2) A2b2 that groups dogs from mainly the Janey B. Goode site in Illinois and Scioto Caverns in Ohio, in addition to older dogs, greater than 4000 years old, from Alabama and Missouri, and (3) a third lineage, A2b3, composed of dogs no older than approximately 4000 years old from throughout the Arctic and along the Pacific coast from Prince Rupert Island, British Columbia to California. In the RAxML trees, PP-00128 occupies two distinct positions depending on the dataset, receiving only poor bootstrap support in all cases. In trees based on datasets 1208 and 1048, it is sister to subclade A2b1, while in trees based on the 1099 and 1106 datasets, PP-00128 is sister to the larger lineage comprising both subclades A2b1 and A2b2. In the BEAST trees, the position of PP-00128 as sister to subclade A2b1 is obtained with datasets 1048 and 1099, whereas dataset 1106 resolves PP-00128 as sister to subclades A2b1 and A2b2, with a posterior probability of 1.0.

It is noteworthy that the phylogenetic position of subclade A2b3 varies greatly among the datasets. In the RAxML analyses, it is sister to subclade A2b2 with the 1048 and 1208 datasets, similarly to the findings of Ní Leathlobhair et al. [11]. On the other hand, based on datasets 1099 and 1106, also including recently published data from Arctic sled dogs [25], subclade A2b3 is sister to the remaining PCD lineage, however, with low bootstrap support. In the BEAST analyses of the 1048 and 1106 datasets, subclade A2b3 is sister to subclade A2b1, although with low posterior probability. With the 1099 dataset, subclade A2b3 is sister to the entire A2 clade, which is strongly supported with a posterior value of 1.0. Subclades A2b1 and A2b2 receive support only in the BEAST trees based on the 1048 and 1099 datasets.

In the mitogenome haplotype network (figure 3), clades A2a and A2b are differentiated by six substitutions. Within clade A2b, PP-00128 is located in the centre of the network, from which the three PCD subclades radiate. PP-00128, which has four mutations that are not shared with any other dog, is closely related to the Koster dogs, separated from them by seven to nine substitutions. All other PCDs are separated from PP-00128 by at least nine substitutions. The three subclades nested inside A2b differ from each other by six substitutions at the most.

## (d) Divergence time and demographic estimations

The divergence time estimates derived from each dataset are comparable. Because it is the most inclusive dataset, estimated divergence times based on the analysis of dataset 1106 are presented in figure 2b (see electronic supplementary material, figure S8 for the full results). Based on the Markov chain Monte Carlo (MCMC) divergence time estimates determined using BEAST for dataset 1106 (figure 2b), the ancestor of precontact and Zhokov dogs lived approximately 16 700 years BP (95% HPD 14 894–18 719 years BP). The last common ancestor of PCDs lived approximately 16 200 years BP (95% HPD 14 378–18 125 years BP), which is also the split date for subclade A2b3. PP-00128 itself split off at approximately 14 500 years BP (95% HPD 12 988–16 261 years BP). Subclades A2b1 and A2b2 split at approximately 14 040 years BP (95% HPD 12 570–15 651 years BP). The Arctic sled dogs that are closely related to Zhokov dogs shared a common ancestor that lived approximately 10 300 years BP (95% HPD 9197–10 998 years BP).

Demographic inference from the ESP option in BEAST (electronic supplementary material figure S11) shows that the precontact dog population coalesced approximately 15 000 years ago. For the initial approximately 5000 years, the population size increased until around 10 000 years ago, after which the population size was constant until approximately 2000 years ago, when a substantial decline initiated.

## 4. Discussion

Due to the close cultural relationship between dogs and humans, and the fact that dogs followed humans to all continents, dogs can be used as a proxy to understand human migration patterns [61]. Before the European colonization of the Americas, dogs entered the New World through Beringia alongside the ancestors of the indigenous peoples of America. However, very few remains from indigenous American dogs have been confirmed and analysed. All PCDs known so far are grouped in a monophyletic clade, A2b, nested inside haplogroup A, indicating a close relationship among all New World indigenous dogs [11]. Even though the oldest human remains and archaeological evidence [5,6,8] are older than any evidence for dogs in the New World, current genetic-based estimates for the initial human migration into the Americas south of the ice sheets [3,4] coincides with the timing reported here for the split of the PCD clade from Eastern Siberian dogs, approximately 16 700 years BP, and the coalescence date for the PCD clade itself,

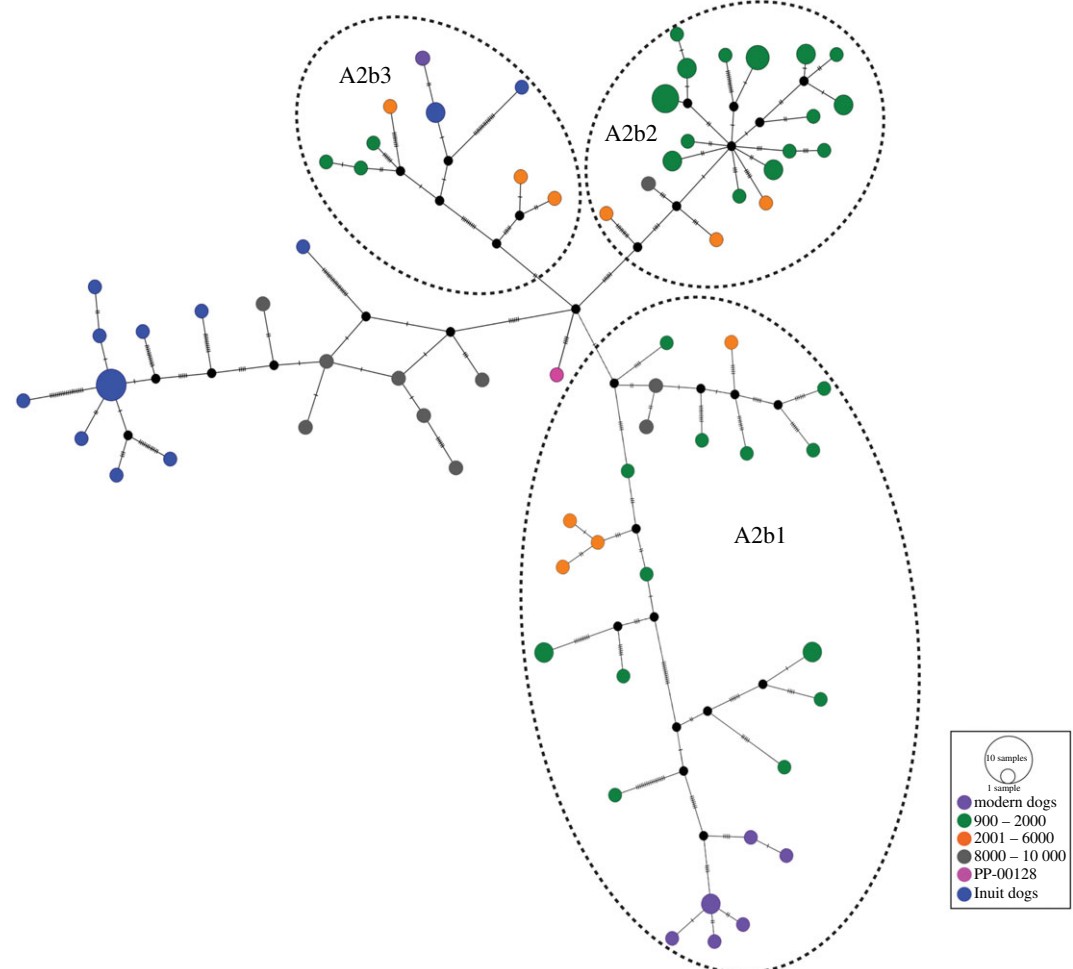

**Figure 3.** Median-joining network of haplogroup A2 dog mitogenome sequences. Circle sizes are proportional to the number of individuals with each haplotype and coloured according to their age group or if they represent Inuit dogs (see their geographical locality in figure 1*b*). (Online version in colour.)

approximately 16 200 years BP. This date is approximately a thousand years older than current estimates [11,61]. While the continental, interior route between the two North American ice sheets only became biologically viable after approximately 13 000 years BP [17,18], the Cordilleran Ice Sheet likely rapidly retreated along the NPC, creating an ecologically viable corridor around southeast Alaska beginning at least 17 000 years ago [20,62], with the inner fjords and sounds ice-free by 15 000 years ago [22]. Archaeological [7] and genetic [3,4] evidence supports the hypothesis that the first humans migrating into the Americas used a coastal route instead of the continental route [5,18,63]. Because the estimated split date of precontact dogs from East Siberian dogs is comparable to the suggested early human migration to North America, it seems likely that dogs were brought with humans during these earliest migrations along the NPC. However, as with humans, physical remains are only known from much later in the subfossil record, and we cannot exclude the possibility that precontact dogs occupied Beringia during the LGM, followed by later migration southward along the coast. The excavated human artefact from Lawyer's Cave that has been dated does not coincide in age with the dog from this cave. Interestingly, though, not far from where PP-00128 was found, 10 300 year-old human remains have been identified from Shuká Káa (also referred to as On Your Knees Cave) on Prince of Wales Island [64,65]. The phylogenetic and geographical position of PP-00128 and this approximately temporally and spatially co-occurring human demonstrates that dogs and humans occupied the

NPC during the Early Holocene, and that at least by this period, a coastal migration route along the NPC was used.

With a mean date of 10 150 cal years BP, the specimen from Lawyer's Cave represents the oldest genetically confirmed dog remains from the Americas, closely followed by a dog from the Stilwell II archaeological site and two dogs from the Koster archaeological site. These mid-continent dogs were of medium size with a significant variation in mandibular size [13]. The limited preservation of the PP-00128 remains (a small fragment of the femur) precludes us from any morphological comparisons with these other early PCDs. In all our phylogenetic analyses, however, the high-coverage mitogenome of PP-00128 is closely related to all PCDs, and in most analyses, sister to a lineage that includes the ancient Stilwell and Koster dogs. Its uncertain, but suggested, basal position among the PCD haplotypes indicates that it belonged to an early lineage of precontact dogs.

Looking at changes in effective population size over time, mitochondrial DNA variation in the PCD population coalesced around 15 000 years BP, after which the population expanded until approximately 10 000 years ago. The population remained at a stable size until 2000 years ago, whereafter it started to decrease until the presumed extinction of the lineage, likely around the time of the European arrival to the North American continent. A previous study found a younger coalescence age of around 9000 years BP, followed by a mostly constant population size [1]. However, this older study sampled a large number of dogs from the same time period and with the same haplotype, and most of the

dog samples were from the Janey B. Goode site, Illinois, likely leading to sampling bias [1]. Similarly to indigenous dogs, the effective population sizes of Native American populations decreased during the same period [1,66]. This decrease coincides with the beginning of the migration of the Inuit Culture [67]. Moreover, during the same period, dogs that inhabited the Arctic before the arrival of the Inuits also experienced a decrease in population size [25]. These results and ours reinforce the notion that dogs may be used as a proxy to analyse human population migration and even demographic patterns, as populations of both species experienced decreases around the same time [1,2,68]. After their arrival to the New World, precontact dogs were isolated until around 2000 years BP. After this time, three new dog introductions to North America occurred, which could have accelerated the population decline of PCDs [11]. Inuits introduced Arctic dogs, also from East Siberia [25,26], European dogs accompanied colonists starting in the 15th century, and most recently, Siberian Huskies were introduced during the Alaska Gold Rush of the twentieth century [28]. When European dog breeds were introduced in America, the PCD population was already decreasing and may have been easily replaced or submersed through genetic swamping. Although European dog breeds and Arctic sled dogs appear to have completely replaced precontact dogs, however, there still appears to be a minor genetic legacy of PCDs among some modern breeds [11,25]. For example, seven modern dogs group inside the PCD clade [11], in addition to two historical dogs [25]. In our analyses, we found another modern dog, an American Eskimo dog from Alaska, grouping with a different PCD subclade from these other modern dogs. These findings corroborate [11] that a higher degree of PCD ancestry, still largely unsampled, may be present among modern or historical American dogs.

Stable isotopes can be used as proxies to infer palaeodiets of organisms. Through stable isotope analysis, it is possible to distinguish if a diet was composed of plants with different photosynthetic pathways (e.g. C3 and C4), and if a diet was based on the consumption of marine versus terrestrial food sources, as marine food and C4 plants tend to have higher concentrations of $^{13}$C when passed to higher levels in the food chain [34,69]. Based on a simple mixture model with cut-offs for each diet type empirically estimated by examining modern diets of species in southeast Alaska, percentage contributions of marine carbon can be hypothesized in the diets of species for which cave bones have been collected [20]. Using this scheme, bones with $\delta^{13}$C values higher than −15‰ are suggested to reflect a diet with a 100% contribution of marine carbon, whereas values between −18‰ and −15‰ indicate a diet with a contribution of 50% of marine carbon, and values lower than −18‰ indicate that diets had no contribution of marine carbon. Moreover, Schoeninger & Tauber [47] state that in historic and prehistoric times, human populations that relied heavily on a marine diet had on average a $\delta^{13}$C value between −12.5‰ and −14.5‰. With a $\delta^{13}$C of −14.1‰, PP-00128 most likely had a mainly marine diet like contemporaneous humans, within the range of marine mammals and similar to diets of ancient dogs from southwest Alaska along the Bering Sea [54]. In Alaska, subsistence-caught fish has been a staple food for sled dogs into recent times, and some types of salmon that are less consumed by people, e.g. the chum salmon (Oncorhynchus keta), also called dog salmon throughout South Alaska [54], may have also been fed to dogs in SE Alaska at the time PP-00128 lived. Leftovers from human hunts (e.g. organs) have also been reportedly used as dog food [70], and such animals could have included subsistence-caught seals and whales in southeast Alaska. Hence, PP-00128 had a marine-based diet, likely similar to humans also occupying the southeast Alaska coast during the Early Holocene.

## 5. Conclusion

The results presented here indicate that PP-00128 represents an early-branching precontact dog, and we suggest that it may be a close relative of the earliest domesticated dogs that accompanied humans during their migrations to the New World. The coastal location of the PP-00128 specimen, combined with its calibrated age and molecular clock estimates for the first New World dogs and human dispersals south of the ice sheets, support that initial human and dog migration occurred together along the NPC route, instead of via the continental interior and subsequent westward movement to the coast. PCDs were almost entirely replaced by later Inuit sled dogs and European breeds that arrived with European colonists, leaving only a minor PCD genetic legacy among modern dogs. Future nuclear genome analyses of precontact dogs, including PP-00128, will permit more complete insights of the fate of PCDs.

Data accessibility. The mitochondrial genome sequence generated in this study is deposited in the NCBI GenBank database with accession no. MW549038. Sequence reads have been deposited in the Sequence Read Archive (SRA) with project number PRJNA694129. The sequence alignment (dataset 1106) is available from the Dryad Digital Repository: https://doi.org/10.5061/dryad.tb2rbp000 [71].

Authors' contributions. C.L. designed the study; F.A.S.C. and S.G. generated the sequence data; F.A.S.C., S.G., C.M.T. and C.L. analysed the data; T.H.H. provided the sample and palaeontological context; F.A.S.C. and C.L. wrote the manuscript with contributions from all authors.

Competing interests. We declare we have no competing interests.

Funding. This research was supported with funding from the National Science Foundation (DEB grant no. 1556565, EAR grant nos. 1854550, 9870343 and 0208247) and the National Geographic Society (grant no. 6212-98).

Acknowledgements. We thank the University of Alaska Museum of the North for loan of the specimen (University of Alaska Museum Earth Sciences Collection, Fairbanks, catalogue number UAMES 52399), Tongass National Forest archaeologists Jane Smith, Gina Esposito and Jackie de Montigny, University of South Dakota student field assistants Frank Andy Klock, Nathan Carter, Brandon Silver, Louis Rezac, Clarissa Ford, Alex Santos and Christy Heaton, and Victor A. Albert for valuable comments on the manuscript.

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
