## [Peer Review File · Proceedings of the Royal Society B: Biological Sciences]

Review History

RSPB-2020-3103.R0 (Original submission)

Review form: Reviewer 1

Recommendation

Accept with minor revision (please list in comments)

Scientific importance: Is the manuscript an original and important contribution to its field?

Good

General interest: Is the paper of sufficient general interest?

Excellent

Quality of the paper: Is the overall quality of the paper suitable?

Good

Is the length of the paper justified?

Yes

Should the paper be seen by a specialist statistical reviewer?

No

Do you have any concerns about statistical analyses in this paper? If so, please specify them explicitly in your report.

No

It is a condition of publication that authors make their supporting data, code and materials available - either as supplementary material or hosted in an external repository. Please rate, if applicable, the supporting data on the following criteria.

Is it accessible?

Yes

Is it clear?

Yes

Is it adequate?

Yes

Do you have any ethical concerns with this paper?

No

Comments to the Author

The authors present the analysis (dating, isotopes, aDNA) of a specimen from coastal Alaska, identified as the earliest domestic dog remains in the Americas. They further use these results to support the hypothesis that humans (and dogs) first colonized the New World via a coastal route. I give my comments here as an archaeologist and do not comment on the specifics of the aDNA analysis, which appears sound.

Firstly, let me say that this is an exciting addition to the American ancient dog record. The unique basal position of the dog, in relation to other precontact dogs, only adds to its importance. As such, the manuscript represents a valuable contribution to the scholarship in this research field and is worthy of publication in ProcB.

One can understand why the authors suggest the finding of this SEAK dog would hint at a coastal migration route, particularly given its more basal position within the PCD subclades, but there is also a possibility that it is the result of a dog population that lingered elsewhere (say, western Alaska) for some period before moving further southward along the coast at a later period. While less likely, this cannot be ruled out as a possibility. A strength of this paper is the use of dogs as a proxy for better understanding human migration.

As an archaeologist I would like to see a more detailed account of the archaeological site from which the specimen was found. The authors offer only scant detail. Given the importance of the find it would be beneficial to give a more thorough account, particularly of the anthropogenic artifacts.

The authors note that PCD subclade 3 consists of Arctic and coastal Alaskan and Canadian dog samples exclusively and that it is the earliest branching PCD subclade at 16.2 ka. This would seem to further support their hypothesis for human and dog migration along the NPC route, but I do not see this mentioned in their conclusions. Surely this would strengthen their argument.

It would be more useful if the circle colors in Figure 1 denoted the PCD subclade of each sample rather than, or in addition to, the time period of the sample. I think readers would be interested in visualizing the geographic spread of the various PCD subclades.

Some additional comments:

Line 103-106: The wording of this sentence is a bit unclear.

Line 110: guessing this is 'as'

Line 119: capitalize Arctic here, as elsewhere

Line 257: Correct to 'Janey B. Goode' and 'Scioto Caverns'

Line 259-260: The authors say there is no association of location with subclade affinity, but is subclade 3 being located only in the coastal Alaska/PNW region not an affinity between location and subclade?

It would be useful to note what the blue circles are in the Figure 1 caption.

Review form: Reviewer 2 (Laurent Frantz)

Recommendation

Accept with minor revision (please list in comments)

Scientific importance: Is the manuscript an original and important contribution to its field?

Excellent

General interest: Is the paper of sufficient general interest?

Excellent

Quality of the paper: Is the overall quality of the paper suitable?

Excellent

Is the length of the paper justified?

Yes

Should the paper be seen by a specialist statistical reviewer?

No

Do you have any concerns about statistical analyses in this paper? If so, please specify them explicitly in your report.

No

It is a condition of publication that authors make their supporting data, code and materials available - either as supplementary material or hosted in an external repository. Please rate, if applicable, the supporting data on the following criteria.

Is it accessible?

Yes

Is it clear?

Yes

Is it adequate?

No

Do you have any ethical concerns with this paper?

No

Comments to the Author

This study provides new and significant insights into both the evolution of dogs and the peopling of the Americas. I found the paper well written, the data properly analysed and the conclusions justified. The fact that the sample sequenced in this study not only is the oldest dog found in the Americas, but also branches out early in the phylogeny is I think very interesting and does

warrant the very balanced discussion about coastal migration in the paper. Overall, I think this paper could be accepted as is - although see minor point below.

The only "major" suggestion I have is that we have a new paper coming out on January 25th in PNAS (see here: <https://doi.org/10.1073/pnas.2010083118>). I am sure that the authors would find this very relevant and although it is not going to change the major conclusion of their study, I think that our findings will be able to provide some interesting background for their discussion particularly about the early peopling of the Americas with respects to dogs.

Minor comments:

Line 47-48: My understanding is that there is evidence of humans much before this - see here: - <https://science.sciencemag.org/content/365/6456/891> - technically this sentence is correct but I feel it is a bit misleading.

Line 190 (and other places): Ní Leathlobhair et al. not "Leathlobhair et al."

I would strongly suggest that the author should submit both the consensus sequence of their new sample but also the reads so that people can integrate this data more easily in their pipeline (e.g. if using assembly instead of consensus).

Laurent Frantz

Decision letter (RSPB-2020-3103.R0)

20-Jan-2021

Dear Professor Lindqvist

I am pleased to inform you that your manuscript RSPB-2020-3103 entitled "An early dog from Southeast Alaska supports a coastal route for the first dog migration into the Americas" has been accepted for publication in Proceedings B. Congratulations!!

The referee(s) have recommended publication, but also suggest some minor revisions to your manuscript. Therefore, I invite you to respond to the referee(s)' comments and revise your manuscript. Because the schedule for publication is very tight, it is a condition of publication that you submit the revised version of your manuscript within 7 days. If you do not think you will be able to meet this date please let us know.

It is a condition of publication that data supporting your paper are made available either in the electronic supplementary material or through an appropriate repository. Please see our Data Sharing Policies <https://royalsociety.org/journals/authors/author-guidelines/#data>.

Sincerely,
Dr John Hutchinson, Editor
mailto: proceedingsb@royalsociety.org

Associate Editor
Board Member: 1
Comments to Author:
Dear Authors,

Thank you for your submission to Proceedings B—this looks to be a very significant paper and it has received strong reviews from two expert reviewers. Both reviewers agree that the paper is close to being acceptable in its present state, although they offer some helpful suggestions that would strengthen the paper further: for example, Referee 1 suggests that providing additional information about the archaeological site would be beneficial, and Referee 2 provides a link to a relevant preprint that may provide additional useful context for this work.

I look forward to seeing your revision, and many thanks again for your submission.

Reviewer(s)' Comments to Author:

Referee: 1

Comments to the Author(s)

The authors present the analysis (dating, isotopes, aDNA) of a specimen from coastal Alaska, identified as the earliest domestic dog remains in the Americas. They further use these results to support the hypothesis that humans (and dogs) first colonized the New World via a coastal route. I give my comments here as an archaeologist and do not comment on the specifics of the aDNA analysis, which appears sound.

Firstly, let me say that this is an exciting addition to the American ancient dog record. The unique basal position of the dog, in relation to other precontact dogs, only adds to its importance. As such, the manuscript represents a valuable contribution to the scholarship in this research field and is worthy of publication in ProcB.

One can understand why the authors suggest the finding of this SEAK dog would hint at a coastal migration route, particularly given its more basal position within the PCD subclades, but there is also a possibility that it is the result of a dog population that lingered elsewhere (say, western Alaska) for some period before moving further southward along the coast at a later period. While less likely, this cannot be ruled out as a possibility. A strength of this paper is the use of dogs as a proxy for better understanding human migration.

As an archaeologist I would like to see a more detailed account of the archaeological site from which the specimen was found. The authors offer only scant detail. Given the importance of the find it would be beneficial to give a more thorough account, particularly of the anthropogenic artifacts.

The authors note that PCD subclade 3 consists of Arctic and coastal Alaskan and Canadian dog samples exclusively and that it is the earliest branching PCD subclade at 16.2 ka. This would seem to further support their hypothesis for human and dog migration along the NPC route, but I do not see this mentioned in their conclusions. Surely this would strengthen their argument.

It would be more useful if the circle colors in Figure 1 denoted the PCD subclade of each sample rather than, or in addition to, the time period of the sample. I think readers would be interested in visualizing the geographic spread of the various PCD subclades.

Some additional comments:

Line 103-106: The wording of this sentence is a bit unclear.

Line 110: guessing this is 'as'

Line 119: capitalize Arctic here, as elsewhere

Line 257: Correct to 'Janey B. Goode' and 'Scioto Caverns'

Line 259-260: The authors say there is no association of location with subclade affinity, but is subclade 3 being located only in the coastal Alaska/PNW region not an affinity between location and subclade?

It would be useful to note what the blue circles are in the Figure 1 caption.

Referee: 2

Comments to the Author(s)

This study provides new and significant insights into both the evolution of dogs and the peopling of the Americas. I found the paper well written, the data properly analysed and the conclusions justified. The fact that the sample sequenced in this study not only is the oldest dog found in the Americas, but also branches out early in the phylogeny is I think very interesting and does warrant the very balanced discussion about coastal migration in the paper. Overall, I think this paper could be accepted as is - although see minor point below.

The only "major" suggestion I have is that we have a new paper coming out on January 25th in PNAS (see here: <https://doi.org/10.1073/pnas.2010083118>). I am sure that the authors would find this very relevant and although it is not going to change the major conclusion of their study, I think that our findings will be able to provide some interesting background for their discussion particularly about the early peopling of the Americas with respects to dogs.

Minor comments:

Line 47-48: My understanding is that there is evidence of humans much before this - see here: - <https://science.sciencemag.org/content/365/6456/891> - technically this sentence is correct but I feel it is a bit misleading.

Line 190 (and other places): Ní Leathlobhair et al. not "Leathlobhair et al."

I would strongly suggest that the author should submit both the consensus sequence of their new sample but also the reads so that people can integrate this data more easily in their pipeline (e.g. if using assembly instead of consensus).

Laurent Frantz

Author's Response to Decision Letter for (RSPB-2020-3103.R0)

See Appendix A.

Decision letter (RSPB-2020-3103.R1)

28-Jan-2021

Dear Professor Lindqvist

I am pleased to inform you that your manuscript entitled "An early dog from Southeast Alaska supports a coastal route for the first dog migration into the Americas" has been accepted for publication in Proceedings B.

Open Access

Paper charges

Sincerely,

Proceedings B

Appendix A

Board Member: 1

Comments to Author:

Dear Authors,

Thank you for your submission to Proceedings B—this looks to be a very significant paper and it has received strong reviews from two expert reviewers. Both reviewers agree that the paper is close to being acceptable in its present state, although they offer some helpful suggestions that would strengthen the paper further: for example, Referee 1 suggests that providing additional information about the archaeological site would be beneficial, and Referee 2 provides a link to a relevant preprint that may provide additional useful context for this work.

I look forward to seeing your revision, and many thanks again for your submission.

We very much appreciate the positive remarks. We have followed all their suggestions and have responded to the reviewer comments point-by-point below.

Reviewer(s)' Comments to Author:

Referee: 1

Comments to the Author(s)

The authors present the analysis (dating, isotopes, aDNA) of a specimen from coastal Alaska, identified as the earliest domestic dog remains in the Americas. They further use these results to support the hypothesis that humans (and dogs) first colonized the New World via a coastal route. I give my comments here as an archaeologist and do not comment on the specifics of the aDNA analysis, which appears sound.

Firstly, let me say that this is an exciting addition to the American ancient dog record. The unique basal position of the dog, in relation to other precontact dogs, only adds to its importance. As such, the manuscript represents a valuable contribution to the scholarship in this research field and is worthy of publication in ProcB.

We thank the reviewer for the positive comments and appreciate the constructive feedback.

One can understand why the authors suggest the finding of this SEAK dog would hint at a coastal migration route, particularly given its more basal position within the PCD subclades, but there is also a possibility that it is the result of a dog population that lingered elsewhere (say, western Alaska) for some period before moving further southward along the coast at a later period. While less likely, this cannot be ruled out as a possibility. A strength of this paper is the use of dogs as a proxy for better understanding human migration.

We agree with the reviewer that we cannot exclude this possibility and we have added a statement about this in the discussion (see first paragraph in Discussion, page 8).

As an archaeologist I would like to see a more detailed account of the archaeological site from which the specimen was found. The authors offer only scant detail. Given the importance of the

find it would be beneficial to give a more thorough account, particularly of the anthropogenic artifacts.

Some of the cave information was left out in the submitted manuscript due to space constraint but we have now added a little more detail of the Lawyer's Cave site. However, other than the two excavations led by Prof. Timothy Heaton in 1998 and 2003, no other work has been done so far at this site and no actual account of this cave had been published nor little is known about the anthropogenic remains and artifacts.

The authors note that PCD subclade 3 consists of Arctic and coastal Alaskan and Canadian dog samples exclusively and that it is the earliest branching PCD subclade at 16.2 ka. This would seem to further support their hypothesis for human and dog migration along the NPC route, but I do not see this mentioned in their conclusions. Surely this would strengthen their argument.

As we note in the paper, the phylogenetic position of subclade 3 is poorly resolved and varies greatly depending on the dataset. Hence, we are hesitant to make any conclusions involving subclade 3 for this study. However, we have described the membership of this clade in a little better detail in the Results.

It would be more useful if the circle colors in Figure 1 denoted the PCD subclade of each sample rather than, or in addition to, the time period of the sample. I think readers would be interested in visualizing the geographic spread of the various PCD subclades.

In addition to the colors denoting the age of the samples, we are now also showing their subclade membership by the shape of the symbols. We find it useful to show both age class and subclade membership on the map.

Some additional comments:

Line 103-106: The wording of this sentence is a bit unclear.

This has been corrected

Line 110: guessing this is 'as'

This has been corrected

Line 119: capitalize Arctic here, as elsewhere

This has been corrected

Line 257: Correct to 'Janey B. Goode' and 'Scioto Caverns'

This has been corrected

Line 259-260: The authors say there is no association of location with subclade affinity, but is subclade 3 being located only in the coastal Alaska/PNW region not an affinity between location and subclade?

As mentioned above, membership of subclade 3 also includes dogs from throughout their North American Arctic distribution, not just the coastal region.

It would be useful to note what the blue circles are in the Figure 1 caption.
The figure legends have been updated to better describe the figure.

Referee: 2

Comments to the Author(s)

This study provides new and significant insights into both the evolution of dogs and the peopling of the Americas. I found the paper well written, the data properly analysed and the conclusions justified. The fact that the sample sequenced in this study not only is the oldest dog found in the Americas, but also branches out early in the phylogeny is I think very interesting and does warrant the very balanced discussion about coastal migration in the paper. Overall, I think this paper could be accepted as is - although see minor point below.

We thank the reviewer for the positive comments and appreciate the constructive feedback.

The only “major” suggestion I have is that we have a new paper coming out on January 25th in PNAS (see here: <https://doi.org/10.1073/pnas.2010083118>). I am sure that the authors would find this very relevant and although it is not going to change the major conclusion of their study, I think that our findings will be able to provide some interesting background for their discussion particularly about the early peopling of the Americas with respects to dogs.

We thank the reviewer for pointing our attention to this new perspective piece. Although it does not present any new data, we do find it highly relevant and have cited the paper in our Discussion.

Minor comments:

Line 47-48: My understanding is that there is evidence of humans much before this - see here: - <https://science.sciencemag.org/content/365/6456/891> - technically this sentence is correct but I feel it is a bit misleading.

We agree with the reviewer and have included the following in this sentence to clarify this: “...evidence from archeological artifacts indicates a presence of human populations south of the continental ice sheets at possibly more than 16,000 years ago”

Line 190 (and other places): Ní Leathlobhair et al. not “Leathlobhair et al.”.

This has been corrected.

I would strongly suggest that the author should submit both the consensus sequence of their new sample but also the reads so that people can integrate this data more easily in their pipeline (e.g. if using assembly instead of consensus).

We agree that it is in the interest of the community that as much data is made publicly available and we have deposited the sequence reads in the NCBI SRA public database, in addition to making the mitogenome sequence available in GenBank.

Laurent Frantz